# Establishing the measurement and psychometrics of medical student feedback literacy (IMPROVE-FL): A research protocol

**Mohamad Nabil Mohd Noor**[1], **Jessica Grace Cockburn**[2], **Chan Choong Foong**[1]*,
**Chiann Ni Thiam**[3], **Yang Faridah Abdul Aziz**[4], **Wei-Han Hong**[1], **Vinod Pallath**[5],
**Jamuna Vadivelu**[1]

1 Medical Education and Research Development Unit, Universiti Malaya, Kuala Lumpur, Malaysia,
2 Department of Surgical Oncology, University Health Network, Toronto, Canada, 3 Department of Internal Medicine, Hospital Pulau Pinang, Pulau Pinang, Malaysia, 4 Department of Biomedical Imaging, Universiti Malaya, Kuala Lumpur, Malaysia, 5 Jeffrey Cheah School of Medicine and Health Sciences, Monash University Malaysia, Selangor, Malaysia

* foongchanchoong@um.edu.my

## Abstract

Current feedback models advocate learner autonomy in seeking, processing, and responding to feedback so that medical students can become feedback-literate. Feedback literacy improves learners' motivation, engagement, and satisfaction, which in turn enhance their competencies. However, there is a lack of an objective method of measuring medical student feedback literacy in the empirical literature. Such an instrument is required to determine the level of feedback literacy amongst medical students and whether they would benefit from an intervention. Therefore, this research protocol addresses the methodology aimed at the development of a comprehensive instrument for medical student feedback literacy, which is divided into three phases, beginning with a systematic review. Available instruments in health profession education will be examined to create an interview protocol to define medical students' feedback literacy from the perspectives of medical students, educators, and patients. A thematic analysis will form the basis for item generation, which will subsequently undergo expert validation and cognitive interviews to establish content validity. Next, we will conduct a national survey to gather evidence of construct validity, internal consistency, hypothesis testing, and test-retest reliability. In the final phase, we will distribute the instrument to other countries in an international survey to assess its cross-cultural validity. This protocol will help develop an instrument that can assist educators in assessing student feedback literacy and evaluating their behavior in terms of managing feedback. Ultimately, educators can identify strengths, and improve communication with students, as well as feedback literacy and the feedback process. In conclusion, this study protocol outlined a systematic, evidence-based methodology to develop a medical student feedback literacy instrument. This study protocol will not only apply to medical and local cultural contexts, but it has the potential for application in other educational disciplines and cross-cultural studies.

**Data Availability Statement:** No datasets were generated or analysed during the current study. All

relevant data from this study will be made available upon study completion.

**Funding:** The work is supported financially by the Ministry of Higher Education Malaysia via Fundamental Research Grant Scheme (FRGS/1/2022/SSI07/UM/02/19) (FP050-2022). The funders had no role in study design, data collection and analysis, decision to publish, or preparation of the manuscript.

**Competing interests:** The authors have declared that no competing interests exist.

# Introduction

The theory of feedback emerged as a form of behaviorism, in which behaviors are reinforced to inform learners [1]. Feedback is an integral part of formative assessments that are a strategy adopted by teachers to direct student learning [2]. Feedback is especially important in medical education as it affects patient safety [3, 4] and clinical competence [5] in addition to scholastic performance [5, 6]. Preparing well-rounded, capable healthcare practitioners so becomes dependent on developing feedback literacy.

The current concept of feedback has since transformed from a unidirectional tool to one involving students' bidirectional engagement for holistic learning [7], which facilitates focused feedback models that target specific cognitive processes [8], student perspectives [9], and the pedagogical aspects of feedback [10]. Accordingly, learners must receive feedback in a positive manner, and teachers need to formulate and deliver feedback efficiently [11]. This shared responsibility motivates students to seek, process, and respond to feedback from multiple sources and different forms [12–15]. Applying external feedback allows them to construct their own internal feedback [16], thereby helping to evaluate their work and decide if they need further guidance [17], which prompts them to seek additional feedback iteratively [14, 15]. Moreover, to refine the feedback process, learners must recognize and communicate their learning requirements [15]. Hence, they are expected to utilize feedback, implement strategies to take action on these suggestions, and improve their competencies [10, 14]. Learners who fulfill these roles are considered feedback-literate [10, 15].

Student feedback literacy refers to the ability to interpret and evaluate received feedback and apply it to enhance competencies [10]. Learners' roles in feedback dialogues are characterized by the features of feedback literacy (Fig 1), which reportedly have cognitive, affective, behavioral, and social benefits for their learning. In terms of cognition, feedback literacy can help learners gain knowledge and appreciate the feedback process [5], thus making learners more likely to engage in feedback activities [18, 19] and coherent in communicating their learning goals with educators [11]. Regarding the affective aspect, learners tend to be more satisfied with the feedback they receive [20] and are more motivated to seek feedback and enact suggestions from these dialogues [19, 21]. Hence, feedback-literate learners display increased confidence in learning from their strengths and weaknesses, thereby advancing their competencies efficiently [6, 21, 22]. In addition, learners' active participation in feedback dialogues can help teachers refine their preparation and delivery of feedback [11]. Therefore, learners' and teachers' feedback literacy levels mutually benefit each other's feedback practices [11]. These tangible benefits highlight the importance of maximizing the feedback process utility.

Measuring feedback literacy can help educators identify areas of strength and improvement to promote the development of feedback-literate learners. Such an instrument would be invaluable in assessing the cycle illustrated in Fig 1 and identifying the specific features and activities with which learners require help. However, a recent scoping review reports that available feedback literacy instruments require further evaluation and validation [23]. Therefore, there is a current need for a valid instrument to be developed based on empirical findings to allow a comprehensive measurement of student feedback literacy. Consequently, interventions for student feedback literacy can be objectively evaluated and compared. In short, a reliable and valid instrument is required to measure student feedback literacy.

When developing a comprehensive instrument for measuring such literacy, several factors must be considered. First, developing such an instrument should include the learners' perspectives as they are integral to effective feedback [10, 15]. However, instruments should be contextualized in the mode of feedback delivery [24]. For example, medical students' feedback literacy should be specialized because it involves patient safety [3] and active workplace

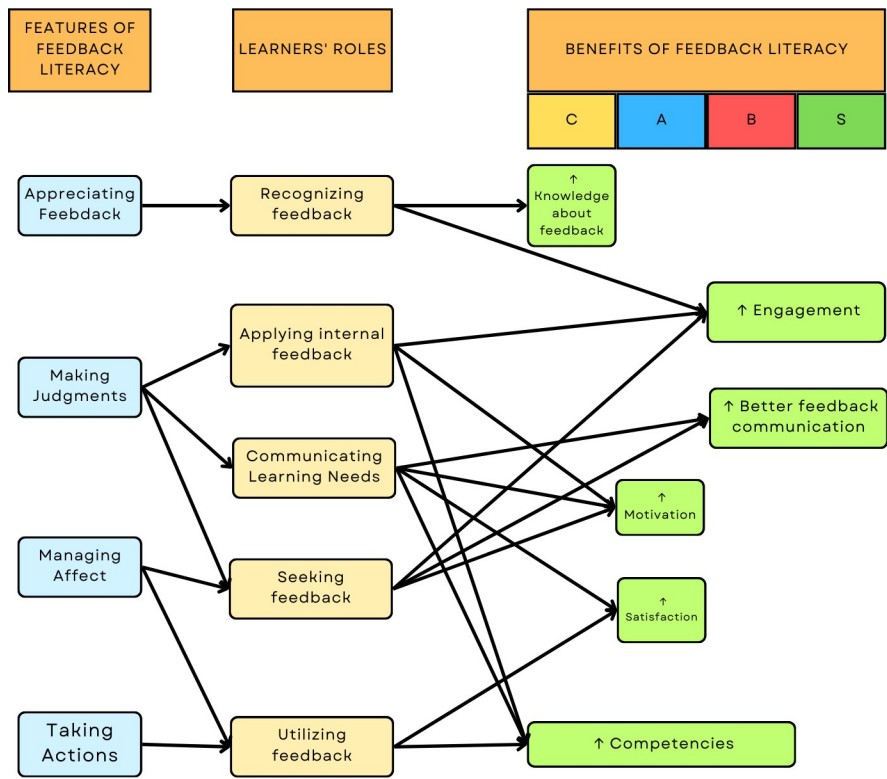

**Fig 1. Feedback literacy, learners' roles, and benefits.** *Notes*: C = Cognitive; A = Affective; B = Behavioral; S = Social.

teaching and learning [25, 26]. By comparison, feedback in language education focuses on written corrective feedback [27, 28], which may not be applicable in the medical education context. Culture also plays a role in feedback practices [29, 30]; hence, the instrument can be examined for both accessibility and inclusiveness. Thus, the methodology for developing a feedback literacy measurement instrument should consider learner perspectives, match the educational or training context, and be culturally sensitive.

Given the complex nature of feedback literacy, this research protocol proposes a methodology that will lead to the development of a comprehensive instrument for medical student feedback literacy. This protocol will serve as a foundation for several related studies to answer the following research questions:

- What are the available instruments for feedback literacy in healthcare education?

- Are these instruments valid and reliable?

- How do medical students, medical teachers and patients conceptualize medical student feedback literacy?

- What are the measurable domains of medical student feedback literacy?

- Are these measurable domains of medical student feedback literacy valid and reliable?

By developing a validated tool to measure feedback literacy, this study aims to fill a critical gap in medical education, ultimately contributing to the improvement of both teaching practices and student outcomes.

## Materials and methods

### Study design

This research project is an exploratory sequential mixed-methods study designed to explore feedback literacy as an educational phenomenon, which in turn becomes the foundation for the development and validation of an instrument [31]. This study will be conducted in three phases (Fig 2).

### Phase 1: Examining measures of medical student feedback literacy

**Systematic review.** Existing feedback literacy instruments in medical education and other health professionals will be reviewed to comprehensively examine the current definitions and metrics used to assess feedback literacy. This review will use guidelines specified in the medical education context [32]. Instruments with similar constructs that measure any features of feedback literacy [10] will be included to provide a deeper understanding of such literacy.

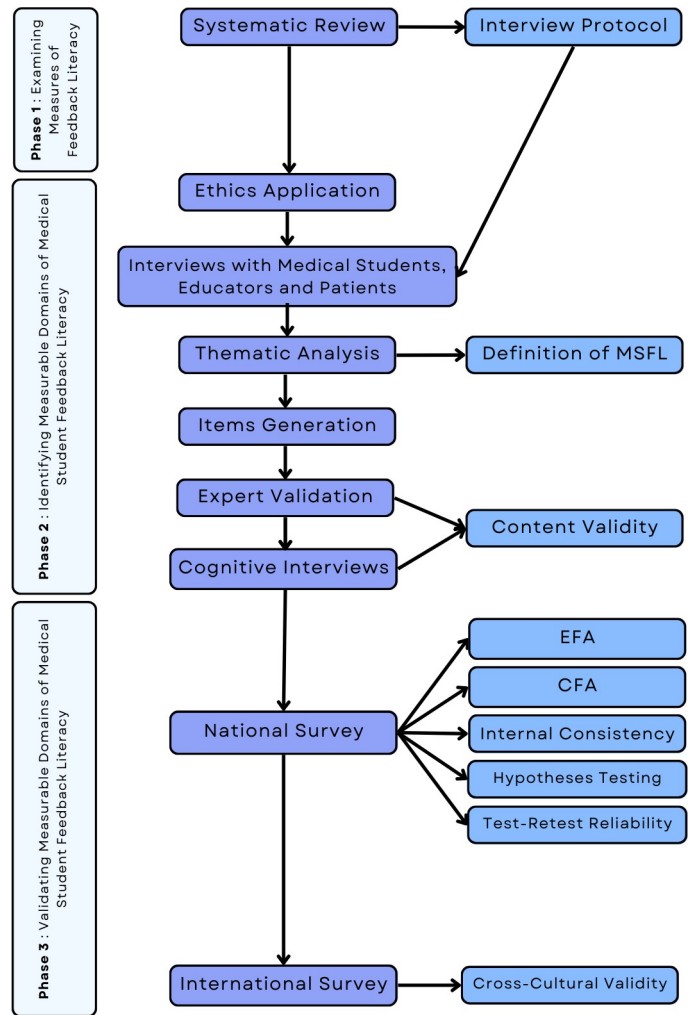

**Fig 2. Flow chart of the research methodology.** *Notes*: MSFL = Medical Student Feedback Literacy, EFA = Exploratory Factor Analysis, CFA = Confirmatory Factor Analysis.

Initially, a pilot search will be performed to select keywords that return relevant research articles. Based on the PICO (Population-Intervention-Comparison-Outcome) format, terms and synonyms related to healthcare education students, feedback literacy, psychometric properties, and instruments will be employed to perform a systematic search of six electronic databases (i.e., Scopus, Medline, Web of Science, CINAHL Complete, Education Research Complete, and Psychology and Behavioral Sciences Collection). The full search string for each database is provided in S1 Appendix. The references and citations of these articles will be manually screened for additional potential articles [32, 33]. Periodic searches will be conducted to identify newly published studies and ensure that the review accounts for all recent publications.

Identified studies will be evaluated using the following set of eligibility criteria:

1. included health professions education of any level (undergraduate and/or postgraduate)

2. instrument measuring student feedback literacy or any of its features (regardless of the feedback provider, e.g. peers, patients, teachers) including adaptations, revalidation in a different context and replication studies

3. focusing on students' role in the feedback process i.e., behaviours and/or attitudes of students towards the feedback process

4. reporting any psychometric properties listed in the Consensus-based Standards for the Selection of Health Measurement Instruments (COSMIN) risk of bias checklist [34]

5. original study published in peer-reviewed journals

6. published in any year

   Studies were further excluded if:

1. the instrument does not emphasize feedback literacy or its specific features (such as combination with other unrelated constructs or only mentions feedback literacy briefly)

2. developed instruments that do not focus on the students' roles in the feedback process, for example, feedback providers opinions on the feedback process

3. non-English publications

4. non-primary research articles, such as reviews, opinion pieces, editorials, or perspectives

5. gray literature (e.g., theses, dissertations, unpublished reports, conference presentations)

6. no available full text

After removing duplicates, two authors will independently screen the retrieved records based on their titles, abstracts, and keywords. Subsequently, the full texts will be screened. Any differing opinions will be resolved through discussions between the authors. Consequently, relevant data, such as the description, utility, constructs, and psychometrics of each included instrument, will be extracted. The methodological quality of the studies will be independently appraised by two authors according to the COSMIN Risk of Bias Checklist [34]. This checklist was selected because of its comprehensive and robust methodology applied in its development. It covers a wide range of psychometric properties thoroughly, including content validity, structural validity, hypotheses testing, cross-cultural validity, internal consistency and test-retest reliability. Findings for the systematic review will be synthesized narratively because it has been successfully used in previous systematic psychometric reviews [35, 36], and it allows analysis between the various conceptualizations of feedback literacy. For example, narrative

synthesis allows flexibility to compare between the concepts of Carless and Boud's feedback literacy [10] and the Bing-You et al. FEEDME Culture [37]. Thus, narrative synthesis will be used for its inclusiveness, unlike other synthesis approach such as meta-analysis, which restricts a synthesis to one single concept used in all studies included. The narrative synthesis will be guided by Popay et al.'s [38] recommendations. Cohen's kappa will be calculated to establish inter-rater reliability during the full-text screening and quality appraisal stages [39, 40]. To ensure ease of communication of findings by complying with a standard, this systematic review will be reported according to the PRISMA guidelines [41]. The outcomes will then guide in developing interview protocols for medical students, educators, and patients.

## Phase 2: Identifying measurable domains of medical student feedback literacy

Ethical approval was granted by the Medical Research Ethics Committee of the University of Malaya Medical Centre (MRECID NO: 20221027–11652). Participant recruitment will follow standard ethical regulations, including voluntary participation, preservation of anonymity, and acquisition of informed consent. Participant recruitment for Phase 2 began on August 11, 2023 and will continue until data collection is completed.

To enhance content validity, the questionnaire development will be guided by Johnson and Christensen [42]. Additionally, we will apply Goh and Blake's [43] recommendation as an example of a feasible and proven methodology in the Malaysian context.

## Interviews with medical students, medical educators, and patients

Semi-structured interviews will be conducted to investigate feedback literacy as an educational phenomenon. This phenomenological approach can provide insights into how medical students experience feedback reception [44]. In addition, as important stakeholders in the provision of feedback [45, 46], educators and patients will be interviewed to derive a comprehensive perspective on the entire feedback process. Focus group discussions with medical students are planned to encourage them to share their experiences [31] and individual interviews will be conducted with medical educators and patients, either face-to-face or virtually.

An interview protocol was designed based on the findings of our review of student feedback literacy conceptualizations [10, 15] and Merriam and Tisdell's four types of good interview questions [47]. For example, "Ideally, what do you think a good medical student should do to make the most of the feedback they receive?" can be used as an "ideal position" question to gather opinions on the feedback literacy feature "taking actions." The interview protocol will be piloted to ensure comprehensiveness and comprehensibility [47]. Any modifications deemed necessary from the pilot test will be applied to the interview protocol to allow flexibility and adaptability. For example, interview questions might need to be revised to include examples to elicit elaborate and accurate responses from the interviewees.

To ensure the quick and wide reach of invitations, interviewees will be recruited via social media posts and student societies. The interviews will include medical students studying in Malaysia, medical educators responsible for providing feedback, and patients with experience in providing feedback to medical students over the past month to reduce the risk of recall bias [48]. Data collection will cease after reaching a point of saturation at which no additional data can be retrieved. The expected number of interviews will include 10 focus group discussions (ideally including four medical students in one group; n = 40), 20 medical educators, and 20 patients or caretakers. Interviewees from various backgrounds will be identified and approached (e.g., ethnicity, year of study/teaching, department, public and private medical school, inpatient, or outpatient) to achieve maximum variation sampling [31]. For example, if

the first interviewee is a Malay, Year 3 medical student from Universiti Malaya doing their orthopaedics rotation, the next interviewee would ideally be a Chinese, Year 5 medical student from Universiti Tunku Abdul Rahman, doing their internal medicine rotation. Interviewees will be recruited to include a most diverse backgrounds possible to achieve the standard of maximum variation sampling.

Researchers who have been specifically trained for this purpose will conduct interviews, which will be audiotaped and then transcribed verbatim for further analysis. The transcripts will be returned to the interviewees for review to ensure accuracy with their experiences (member-checking) [31].

**Thematic analysis.**    Interview transcripts will be thematically analyzed using NVivo as this approach is flexible and accessible [49]. As an example, medical students, educators, and patients will be asked the prompt, "Please describe how medical students should receive feedback." Accordingly, triangulating the different experiences between three types of interviewees (source triangulation) may provide a comprehensive description of medical student feedback literacy. Codes will be developed based on the collected data [31, 49] by one researcher. This inductive approach allows for a novel and comprehensive description of medical students' feedback literacy that is not confined to existing concepts. This researcher will read the interview transcripts multiple times to become familiar with them. Recurring phrases or keywords will be highlighted to identify possible patterns in the data and cross-check those that appear among medical students, educators, and patients. Subsequently, codes will be generated to capture notable patterns, which will be checked to reduce overlap and redundancy, and subsequently grouped into potential themes using an iterative approach to refine each theme. These themes will be named and clearly defined. Next, these will be examined across the entire dataset to ensure that the conclusions drawn are firmly rooted in the data. The themes will be organized and presented as a thematic map. Additionally, at least two researchers (other than the coder) will review the themes to enhance the credibility of the findings [50]. Finally, the themes will be analyzed to generate measurable domains of medical students' feedback literacy.

## Phase 3: Validating the measurable domains of medical student feedback literacy

**Item generation.**    The systematic review and thematic analysis of the interviews will be used to produce a list of items. To ensure clarity and enhance the content validity, these will be based on the recommendations from previous research [42, 51, 52]. The words used to develop the items will be guided by the interviewees as much as possible to avoid overinterpretation and to ensure that the items represent the intact ideas of medical students, educators, and patients. Reverse-worded items will be included to minimize response sets [42].

For instance, thematic analysis may ultimately suggest modifying the general item "I have realized that feedback from other people can enhance my self-reflection on how I can systematically improve my learning" [53] to suit the medical education context to "I realize that feedback from supervisors enables me to reflect on how I can improve my clinical skills."

Thematic analysis may also suggest using a new item to suit the local culture, such as, "I accept criticism on my weaknesses that considers the ways (e.g., religion) I was brought up" or "I accept criticism on my weaknesses that is not delivered in front of my peers."

Participants will be asked to rate their responses using a 7-point semantic differential scale ranging from 1 (*do not correspond at all*) to 7 (*correspond exactly*). This 7-point semantic differential scale is chosen to improve the reliability of the instrument (as opposed to a 5 or 6-point scale) [54] while maintaining conciseness (as opposed to an 8, 9 or 10-point scale) [42], and allow a participant to express a true neutral stance (as opposed to an even-numbered

scale) [42]. The Flesch-Kincaid reading ease score will be examined to ensure the items are clear and concise for easier interpretation [55, 56].

**Expert validation.**    To guarantee the item's relevance and representation of the target construct, expert validation will be performed according to the best practices recommended by Yusoff [57]. A panel of experts will review the list of generated items after they are provided with instructions, definitions, and ratings to facilitate the evaluation process. A target number of ten experts will be recruited to assist in this step because the minimum recommended number is six [57], which will account for the possibility of dropouts. A total of four types of experts will be included. They will consist of clinicians who represent the end users of the medical training's outcomes. Experts representing those in cultural sensitivity in Malaysian society will also help vet sensitivity, accessibility, and inclusivity. We will include experts in feedback literacy and questionnaire development.

The expert validation form will be distributed via email. The panel of experts will be asked to evaluate the items and provide comments to improve the instrument before independently rating the items. If needed, virtual or physical meetings will be conducted for additional consultation and explanations. Based on their scores on the content validation index (CVI), items will be refined or discarded. With ten experts, a CVI score of at least 0.78 should be achieved to ensure content validity [58].

**Cognitive interviews.**    The first draft of the questionnaire will be validated by experts and then tested with a group of medical students to strengthen comprehensibility. Cognitive interviews allow the exploration of the reasons why medical students might find the questionnaire to be comprehensible or lack thereof. The execution of cognitive interviews will follow the recommendations of Scott et al., who provide a step-by-step approach with illustrative examples [59]. Semi-structured individual interviews will be conducted with Malaysian medical students, who are the target population of the survey, face-to-face or virtually, and a researcher will be trained to conduct them. The interviews will be audiotaped and then transcribed verbatim.

An interview protocol will be designed based on the drafted items and piloted with two to three medical students to test its feasibility and implementation of the interview protocol [60]. The cognitive interview protocol might be revised after the pilot test to ensure that the questions elicit the needed response from medical students regarding the comprehensibility of the questionnaire items. During the interviews, the think-aloud approach will be adopted to ask interviewees to verbalize their thought processes while answering the questionnaires [61]. In addition, probing questions will be used to guide the focus on specific parts and ensure clarity among the interviewees [62]. Interview transcripts will be analyzed to identify areas for improvement and the questionnaire modified based on these findings. The cognitive interviews will be repeated until no new problems are identified and the questionnaire is deemed comprehensible and comprehensive [62].

**National survey.**    To extend its accessibility to participants nationwide, the medical student feedback literacy scale will be administered online. Approximately 3000 medical students are studying in Malaysia [63]. According to Krejcie and Morgan's [64] calculations, the minimum sample size is 341. An online sample size calculator (http://www.raosoft.com/samplesize.html) (5% margin of error, 95% confidence interval) verified that a minimum of 341 participants is required. To accommodate the planned factor analysis, a larger sample size may be required to achieve a 7:1 ratio of items to participants, depending on the number of items developed [34]. Medical students in Malaysia will be recruited via social media posts and student societies. Recruitment will be periodically performed until a sufficient sample size is obtained.

This national survey will ask participants about their age, sex, ethnicity, institution, and year of study. The data will demonstrate whether the sample reflects different backgrounds to

represent the Malaysian population. Recruitment will be repeated until the sample accounts for the various backgrounds. The researchers will be prepared to recruit a larger sample to guarantee representation.

The adequacy of the sample size should also fulfill the requirements of statistical analyses. This protocol will follow a generic recommendation that at least 200 respondents are required [65]. In addition, for exploratory factor analysis (EFA), the recommended respondent-to-item ratio is 1:10 [66], or a minimum of 1:5 [67, 68]. For confirmatory factor analysis (CFA), the recommended respondent-to-item ratio is 1:10 [69].

Data collected from the national survey will be analyzed for construct validity, internal consistency, hypothesis testing, and test-retest reliability. All data will be managed and analyzed using IBM SPSS Statistics 28.

**Construct, convergent, and discriminant validity.** To guarantee a comparable and representative sample, data for EFA and CFA will be collected from the same national survey. The samples will be randomly split into two subsamples prior to data analysis to undergo either EFA or CFA [70].

The EFA will be performed to establish a hypothesis for the factor model among Malaysian medical students. Furthermore, EFA will be performed according to the guidelines described by Hair et al. [71]. A principal component analysis (PCA) with varimax rotation will be performed. The Bartlett's test measure of sampling adequacy should be significant at $p = 0.05$, and the Kaiser-Meyer-Olkin (KMO) should be above 0.7 [72]. The initial PCA will yield the proposed factors. Those with eigenvalues greater than 1 in the scree plot will be considered [73]. Next, items with an average communality above 0.6 [74] will be considered. In addition, items with a factor loading above 0.5 in one factor [71] and without cross-loading in another factor above 0.5 [75] will be retained for subsequent analyses.

To further assess whether the hypothesis proposed by the EFA is plausible, the items retained from the EFA will undergo CFA. The CFA will be performed according to the guidelines developed by Hair et al. [71]. Indices such as the ratio of chi-square and degrees of freedom, goodness-of-fit index (GFI), Tucker-Lewis index (TLI), comparative fix index (CFI), adjusted goodness-of-fit index (AGFI), and root mean square error of approximation (RMSEA) will be evaluated. A model will be deemed to have a good fit if the chi-square to degrees of freedom ratio is less than three, the RMSEA is less than 0.08, and the other indices have values greater than 0.9 [71]. Items not meeting these criteria will be excluded from subsequent analyses.

In addition, the structure of the model will be assessed for convergent (i.e., the extent to which an indicator of a specific construct converges) and discriminant validity (i.e., distinct constructs are truly unique and measure phenomena that other constructs do not). To achieve convergent validity, the factor loadings and average variance extracted (AVE) and construct reliability should be equal to or higher than 0.5 and 0.7, respectively [71]. No cross loadings should occur between indicators to support discriminant validity, and the AVE values should be higher than the squared correlation estimates [71].

**Internal consistency.** Based on the factors produced via the EFA, the internal consistency between these factors will be determined to evaluate whether the scores are consistent throughout the instrument [31]. To guarantee internal consistency, a Cronbach's alpha of more than 0.70 will be targeted [71]. Items should demonstrate a coefficient above 0.5 and have a corrected item-total correlation above 0.2 within a factor [72, 76].

**Hypotheses testing.** The Implicit Theory of Intelligence Scale [77] will be administered simultaneously during the national survey. The purpose is to further support the construct validity of the instrument through hypothesis testing. Such testing will support the structure of a new construct by correlating it with a similar construct [34].

Dweck's Mindset Theory [78] postulates that individuals' mindsets range from a growth to fixed mindset. People with the former believe they can enhance their abilities by receiving and applying feedback while those with the latter tend to believe that their abilities are predetermined. The proposed hypotheses are that a higher level of feedback literacy will be correlated with a growth mindset, and a lower level with a fixed mindset. Pearson's correlation coefficient higher than 0.70 will be targeted to establish a strong correlation between the two constructs [79].

**Test-retest reliability.**   Test-retest reliability will be evaluated to examine the stability of our measurements over time [31]. Medical students at the Universiti Malaya will be recruited to complete the instrument twice. A sample of 50 medical students will be used to obtain heterogeneity within the sample [80]. To prevent recall and minimize the effect of behavioral changes on feedback literacy, the time interval will be two weeks [81, 82]. Students will be asked to fill in their student identification numbers to match their responses; however, to preserve their anonymity, these will be replaced by running numbers. Based on the McGraw and Wong convention, the reliability score will be calculated with a two-way mixed effect, single measurement, absolute agreement intraclass coefficient (ICC) [80, 83]. A minimum value of 0.75 will be targeted to achieve good test-retest reliability [80, 84].

**International survey.**   After the national survey is completed, the questionnaire will be disseminated online to medical students from other countries to gather evidence of its cross-cultural validity. According to Hofstede's cultural dimensions, the survey will be replicated in countries with similar dimensions [85] (e.g., Singapore) and dissimilar (e.g., Canada) to Malaysia to allow comparison with a range of cultural contexts. We will apply for ethical approval at the respective institutions, and the sample size will be determined according to the number of medical students in the country and to achieve a 1:10 ratio of items to participants for factor analysis.

**Cross-cultural validity.**   Cross-cultural validity will be investigated using multi-group CFA (MGCFA). This method has been chosen for its ability to assess all hierarchical factorial invariances and both uniform and nonuniform item biases [86]. MGCFA will be performed following Gregorich's recommendations [87]. Dimensional, configural, metric, and scalar variances will be examined to support cross-cultural validity. Dimensional and configural invariance will be determined by comparing the number of common factors and their associated items, respectively [87]. Metric invariance will be investigated by fixing equality constraints on the matching factor loadings and concurrently fitting the factor model to sample data from each group [87]. Scalar invariance will be assessed by fixing equality constraints on the corresponding factor loadings and item intercepts and fitting the common factor model to the sample data from each group simultaneously [87]. Differences in the fit indices of 0.01 or less support the metric and scalar invariance hypotheses, respectively [86]. If the metric invariance hypothesis is supported, comparisons of correlations and mean patterns are valid across cultural samples. Meanwhile, a supported scalar invariance hypothesis allows for the comparison of instrument scores across cultural samples.

If different cultural models exhibit partial factorial invariance or non-invariance, items and factors that do not fit the invariance criteria will be analyzed to determine their causes. Misfit items, factors, and samples will be excluded to assess whether this results in improved invariance indicators [86]. Based on the results, the models will be assessed for cultural construct bias and extreme response styles across the samples [86, 87].

## Discussion

Feedback literacy empowers learners to play active roles in feedback processes and improves learners' engagement [18, 19], satisfaction [20], and motivation [19, 21] and, in turn, enhances

their competencies [6, 21, 22]. Thus, a measurement of feedback literacy is needed to allow educators to improve their understanding of learners' attitudes and behaviors when receiving feedback. Consequently, educators can capitalize on this information by identifying learners' strengths and weaknesses in terms of receiving feedback.

An empirical study following this protocol is expected to produce a valid and reliable instrument for measuring medical students' feedback literacy. Furthermore, this study is expected to consolidate the perspectives of feedback providers (i.e., medical educators and patients) and users (i.e., medical students) to derive a learner-inclusive definition of feedback literacy. In addition, the measurement will be specialized based on the educational context (i.e., medical education), and cultural sensitivity will be considered. As feedback practices are influenced by sociocultural and learning environments, this instrument should provide a comprehensive understanding of learners' attitudes and behaviors toward receiving feedback.

The methodology presented in this protocol has several strengths. This research study can explore feedback literacy in medical education to create a consolidated definition. Each step of the project will be guided by validated and frequently used methods for producing an evidence-based research protocol. The instrument will undergo multiple procedures to ensure its validity and reliability, including expert validation and cognitive interviews for content validity, EFA and CFA for construct, convergent, and discriminant validities, calculation of Cronbach's alpha for internal consistency, hypothesis testing with the Implicit Theory of Intelligence Scale for construct validity, test-retest reliability, and international surveys for cross-cultural validity. Implementation of this protocol should result in a comprehensive and accurate measurement of medical students' feedback literacy.

Researchers expect differences in opinion regarding feedback literacy among various stakeholders. For instance, medical educators may have higher expectations from students in their roles as feedback receivers. Such varying perspectives may require a thorough triangulation of their views and a comparison with the literature to generate a consensus. Consideration of the cultural complexities of the research location (i.e., Malaysia) is critical. Therefore, careful verification with local cultural experts is being planned for this study. If not approached appropriately, these challenges may convolute the study results.

Several technical factors must be considered before implementing this protocol, which is being guided by research articles that recommend EFA and CFA to establish construct validity [34, 71]. Other methods, such as Rasch analysis, are also appropriate for instrument validation. Similar to other surveys, this study will be subject to self-reporting bias [48, 88]. The instrument being developed will depend on the reflections and knowledge of feedback literacy among the respondents. This study is also limited by its cross-sectional design, which exposes it to confounding variables [89, 90], such as different learning environments and teacher feedback literacy. Therefore, the next step will be conducting a longitudinal study to assess the development of student feedback literacy.

The feedback literacy scale also has other potential applications. In addition to being a self-assessment tool for medical students, educators can use it to assess students' feedback literacy. Accordingly, they could evaluate their students' behavior in terms of receiving feedback. Ultimately, educators can identify areas of strength, improve communication with students, and enhance feedback literacy and the feedback process. Next, despite the systematic approach to developing and validating feedback literacy (i.e., the parameter) for medical students in Malaysia, the methodology outlined herein could prompt other researchers to adapt it to their parameters and cultural contexts when developing an instrument.

## Conclusion

In conclusion, this study protocol outlined a systematic, evidence-based methodology to develop a medical student feedback literacy instrument. Careful consideration was given into the future development of this instrument so that it will have good internal consistency, test-retest reliability, content validity, construct validity, hypotheses testing and cross-cultural validity. The methodology outlined here does not only serve medical and local cultural contexts, but it has the potential for application in other educational disciplines and cross-cultural studies. Moving forward, researchers can use this study protocol as a guide to develop instruments, measure, and design effective interventions for student feedback literacy.

## Supporting information

**S1 Appendix. Full search string of each database.**
(DOCX)

## Acknowledgments

The research project in this protocol will be conducted as part of fulfilling the requirements of a doctoral degree at Universiti Malaya.

## Author Contributions

**Conceptualization:** Mohamad Nabil Mohd Noor, Jessica Grace Cockburn, Chan Choong Foong, Chiann Ni Thiam, Yang Faridah Abdul Aziz, Wei-Han Hong, Vinod Pallath, Jamuna Vadivelu.

**Data curation:** Mohamad Nabil Mohd Noor, Jessica Grace Cockburn, Chan Choong Foong, Chiann Ni Thiam.

**Formal analysis:** Mohamad Nabil Mohd Noor, Jessica Grace Cockburn, Chan Choong Foong.

**Funding acquisition:** Mohamad Nabil Mohd Noor, Chan Choong Foong, Chiann Ni Thiam.

**Investigation:** Mohamad Nabil Mohd Noor, Jessica Grace Cockburn, Chan Choong Foong, Chiann Ni Thiam, Yang Faridah Abdul Aziz, Wei-Han Hong, Vinod Pallath.

**Methodology:** Mohamad Nabil Mohd Noor, Jessica Grace Cockburn, Chan Choong Foong, Chiann Ni Thiam, Yang Faridah Abdul Aziz, Wei-Han Hong, Vinod Pallath, Jamuna Vadivelu.

**Project administration:** Mohamad Nabil Mohd Noor, Jessica Grace Cockburn, Chan Choong Foong, Chiann Ni Thiam, Yang Faridah Abdul Aziz, Wei-Han Hong, Vinod Pallath.

**Resources:** Mohamad Nabil Mohd Noor, Jessica Grace Cockburn, Chan Choong Foong.

**Software:** Mohamad Nabil Mohd Noor, Chan Choong Foong.

**Supervision:** Jessica Grace Cockburn, Chan Choong Foong, Jamuna Vadivelu.

**Validation:** Mohamad Nabil Mohd Noor, Jessica Grace Cockburn, Chan Choong Foong.

**Visualization:** Mohamad Nabil Mohd Noor, Chan Choong Foong.

**Writing – original draft:** Mohamad Nabil Mohd Noor, Jessica Grace Cockburn, Chan Choong Foong.

**Writing – review & editing:** Mohamad Nabil Mohd Noor, Jessica Grace Cockburn, Chan Choong Foong, Chiann Ni Thiam, Yang Faridah Abdul Aziz, Wei-Han Hong, Vinod Pallath, Jamuna Vadivelu.

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
