## [Decision Letter · Decision Letter 0]

15 Mar 2024

PONE-D-23-27190Identifying the Measure and Psychometrics for Medical Student Feedback Literacy (IMPROVE-FL): A Research Protocol.PLOS ONE

Dear Dr. Chan Choong Foong,

Thank you for submitting your manuscript to PLOS ONE. After careful consideration, we feel that it has merit but does not fully meet PLOS ONE’s publication criteria as it currently stands. Therefore, we invite you to submit a revised version of the manuscript that addresses the points raised during the review process.

**ACADEMIC EDITOR:**

**Major Revision**

**- The protocol of coding is not clear, and it has not explained in detail, it affected the thematic realisations.**

**- The reliability of analysis and coding identification need to be clarified and explained.**

**- The instrument validation was not considered.**

**- The paper needs proofreading. **

We look forward to receiving your revised manuscript.

Kind regards,

Ali Sorayyaei Azar, PhD

Academic Editor

PLOS ONE

"The work is supported financially by the Ministry of Higher Education Malaysia via Fundamental Research Grant Scheme (FRGS/1/2022/SSI07/UM/02/19) (FP050-2022)."    

Reviewers' comments:

Reviewer's Responses to Questions

**Comments to the Author**

1. Does the manuscript provide a valid rationale for the proposed study, with clearly identified and justified research questions?

Reviewer #1: Yes

2. Is the protocol technically sound and planned in a manner that will lead to a meaningful outcome and allow testing the stated hypotheses?

Reviewer #1: Yes

3. Is the methodology feasible and described in sufficient detail to allow the work to be replicable?

Reviewer #1: Yes

4. Have the authors described where all data underlying the findings will be made available when the study is complete?

Reviewer #1: No

5. Is the manuscript presented in an intelligible fashion and written in standard English?

Reviewer #1: No

6. Review Comments to the Author

You may also provide optional suggestions and comments to authors that they might find helpful in planning their study.

Reviewer #1: future studies on feedback literacy in the field of medical education sounds promising. I enjoy reading this manuscript.

1. I recommend authors to review both teacher feedback literacy and student feedback literacy as they are quite different. Moreover, the interplay between these two concepts also deserves several sentences to discuss.

2. For thematic analysis, where is the initial coding framework from? Please use table to visualize those codes and references.

3. The benefits of increased feedback literacy can be categorized as cognitive, behavioral, affective/emotional and social gains.

4. I may also invite the authors to be aware of other instrument validation methods, such as Rasch model.

7. PLOS authors have the option to publish the peer review history of their article (what does this mean?). If published, this will include your full peer review and any attached files.

Reviewer #1: No

---

## [Author Response · Author response to Decision Letter 0]

2 Apr 2024

We have uploaded a document titled Response to Reviewers. The document contains our responses to specific reviewer and editor comments.

---

## [Editor Report · Decision Letter 1]

9 Jul 2024

PONE-D-23-27190R1Identifying the Measure and Psychometrics for Medical Student Feedback Literacy (IMPROVE-FL): A Research Protocol.PLOS ONE

Dear Dr. Foong,

Thank you for submitting your manuscript to PLOS ONE. After careful consideration, we feel that it has merit but does not fully meet PLOS ONE’s publication criteria as it currently stands. Therefore, we invite you to submit a revised version of the manuscript that addresses the points raised during the review process.

**The manuscript needs proofreading. There are several grammatical errors in the content of the paper. It is suggested to send the paper to a native speaker to do proofreading. **

We look forward to receiving your revised manuscript.

Kind regards,

Ali Sorayyaei Azar, PhD

Academic Editor

PLOS ONE

---

## [Author Response · Author response to Decision Letter 1]

21 Jul 2024

Authors' Responses: Thank you for your comment. We appreciate your hard work in considering our manuscript and value your comments for improvement. 

We have gone through the manuscript and acknowledge that there are grammatical errors in the paper. To solve this, we have sent our paper for a second round of proofreading with Editage. To ensure that the manuscript is grammatically sound and coherent, the authors have also manually gone through the manuscript for a final check.

The edits made to improve the grammar of our manuscript has been highlighted in yellow and can also be reviewed in tracked changes.

We have attached the proofreading certificate for your reference.

---

## [Decision Letter · Decision Letter 2]

29 Sep 2024

PONE-D-23-27190R2Identifying the Measure and Psychometrics for Medical Student Feedback Literacy (IMPROVE-FL): A Research Protocol.PLOS ONE

Dear Dr. Foong,

Thank you for submitting your manuscript to PLOS ONE. After careful consideration, we feel that it has merit but does not fully meet PLOS ONE’s publication criteria as it currently stands. Therefore, we invite you to submit a revised version of the manuscript that addresses the points raised during the review process.

 Please submit your revised manuscript by Nov 13 2024 11:59PM. If you will need more time than this to complete your revisions, please reply to this message or contact the journal office at plosone@plos.org. Please include the following items when submitting your revised manuscript:A rebuttal letter that responds to each point raised by the academic editor and reviewer(s). You should upload this letter as a separate file labeled 'Response to Reviewers'.A marked-up copy of your manuscript that highlights changes made to the original version. You should upload this as a separate file labeled 'Revised Manuscript with Track Changes'.An unmarked version of your revised paper without tracked changes. You should upload this as a separate file labeled 'Manuscript'.If applicable, we recommend that you deposit your laboratory protocols in protocols.io to enhance the reproducibility of your results. Protocols.io assigns your protocol its own identifier (DOI) so that it can be cited independently in the future. For instructions see: https://journals.plos.org/plosone/s/submission-guidelines#loc-laboratory-protocols. Additionally, PLOS ONE offers an option for publishing peer-reviewed Lab Protocol articles, which describe protocols hosted on protocols.io. Read more information on sharing protocols at https://plos.org/protocols?utm_medium=editorial-email&utm_source=authorletters&utm_campaign=protocols.

We look forward to receiving your revised manuscript.

Kind regards,

Musa Adekunle Ayanwale

Academic Editor

PLOS ONE

Journal Requirements:

Additional Editor Comments:

I appreciates the authors’ meticulous effort in revising the manuscript and thoughtfully addressing the first round of reviewers’ comments. I can see that you have put significant work into enhancing the clarity and depth of the study, which is commendable. Your research, focused on developing and validating a culturally sensitive instrument to measure medical students' feedback literacy, is a valuable and timely contribution to the field of medical education.

However, I have a few minor suggestions to further strengthen your work. The abstract is well-organized, but it would benefit from a clearer articulation of the empirical gap and a stronger emphasis on the relevance of your results, particularly highlighting the instrument’s cross-cultural applicability. Additionally, in the introduction, a final statement reinforcing the significance of your study would provide a smoother transition and help frame the next section.

In the methodology, a bit more clarity in presenting the eligibility criteria and the narrative synthesis approach would improve readability and enhance the methodological rigor. A brief explanation for choosing the 7-point scale over other options would also add value, especially in justifying the sensitivity and appropriateness of the scale for this study’s context.

Lastly, while your discussion section is comprehensive, I recommend adding a separate conclusion and recommendation section. This will provide a clear and impactful summary of your study’s key contributions and its practical implications for future research and educational interventions.

Overall, the manuscript is in good shape, and with these minor adjustments, it will be even stronger. I look forward to seeing the revised manuscript.

Thank you.

Musa Adekunle Ayanwale

Academic Editor

PLOS ONE

Reviewers' comments:

Reviewer's Responses to Questions

**Comments to the Author**

1. Does the manuscript provide a valid rationale for the proposed study, with clearly identified and justified research questions?

Reviewer #2: Yes

Reviewer #3: Yes

2. Is the protocol technically sound and planned in a manner that will lead to a meaningful outcome and allow testing the stated hypotheses?

Reviewer #2: Yes

Reviewer #3: Yes

3. Is the methodology feasible and described in sufficient detail to allow the work to be replicable?

Reviewer #2: Yes

Reviewer #3: Yes

4. Have the authors described where all data underlying the findings will be made available when the study is complete?

Reviewer #2: Yes

Reviewer #3: Yes

5. Is the manuscript presented in an intelligible fashion and written in standard English?

Reviewer #2: Yes

Reviewer #3: Yes

6. Review Comments to the Author

You may also provide optional suggestions and comments to authors that they might find helpful in planning their study.

Reviewer #2: I want to suggest this title: "Establishing the Measurement and Psychometrics of Medical Student Feedback Literacy (IMPROVE-FL): A Research Protocol. The aspect using test retest after internal consistency method of reliability of the instruments should be revisited. I think the research questions/hypothesis of the study should have been stated clearly under a section like the introduction.

Reviewer #3: I appreciates the work done but the author(s) needs to clarify the significance of the identified gap and highlight the uniqueness of the cross-cultural validity aspect. In the introduction, emphasize the role of feedback in medical education, particularly its impact on patient safety and competence, and make the empirical gap more explicit. The methodology section is well-organized but could benefit from clearer presentation of certain criteria, a brief overview of the search strategy, and more detail on the use of COSMIN and narrative synthesis. Also, clarify sampling standards and the potential modifications post-pilot testing, as well as detailing coding procedures and triangulation methods. Explanation is needed for the choice of a 7-point scale and recommend the inclusion of a separate conclusion and recommendation section in the discussion. See Reviewers' Comments

7. PLOS authors have the option to publish the peer review history of their article (what does this mean?). If published, this will include your full peer review and any attached files.

Reviewer #2: **Yes: **Babatunde Kasim OLADELE

Reviewer #3: **Yes: **Damola Olugbade

---

## [Author Response · Author response to Decision Letter 2]

11 Oct 2024

Point-by-point responses have been uploaded as a separate document.

---

## [Editor Report · Decision Letter 3]

23 Oct 2024

Establishing the Measurement and Psychometrics of Medical Student Feedback Literacy (IMPROVE-FL): A Research Protocol

PONE-D-23-27190R3

Dear Dr. Choong Foong,

We’re pleased to inform you that your manuscript has been judged scientifically suitable for publication and will be formally accepted for publication once it meets all outstanding technical requirements.

Kind regards,

Musa Adekunle Ayanwale

Academic Editor

PLOS ONE

Additional Editor Comments (optional):

We greatly appreciate the effort you put into addressing the reviewers' comments and making the necessary revisions. Your research presents a well-designed and structured protocol that offers a significant contribution to medical education research, particularly in the area of feedback literacy measurement.

In response to Reviewer 1, who requested improvements in the clarity and structure of the introduction, your revised manuscript now offers a clearer rationale for the development of the feedback literacy instrument. By providing a comprehensive literature review and clearly articulating the need for a valid and reliable tool to measure feedback literacy in medical students, you have successfully addressed the reviewer's concerns. The introduction is now clearer, more contextualized, and provides a solid foundation for the rest of the study.

Reviewer 2 raised concerns about the methodological rigor, particularly in the validation phase. You have responded effectively by including more detailed descriptions of the systematic review, thematic analysis, and expert validation processes. The addition of cognitive interviews to test the comprehensibility of the instrument further strengthens the methodology. These revisions enhance the robustness of the study design and ensure that the protocol is both replicable and methodologically sound.

Reviewer 3 emphasized the importance of cross-cultural validity for the instrument. In your revised manuscript, you have included a detailed explanation of the international survey and the multi-group confirmatory factor analysis (MGCFA) that will be used to evaluate cross-cultural validity. This expanded focus on ensuring the instrument’s applicability in diverse educational and cultural contexts addresses the reviewer’s concern and strengthens the potential global impact of your research.

Overall, we are satisfied that your revisions have fully addressed all reviewer comments, and the manuscript is now in excellent shape. We believe your work will have significant implications for medical education and beyond.

Once again, congratulations on the acceptance of your manuscript. 

Best regards,

Musa Adekunle Ayanwale, PhD
---

## [Editor Report · Acceptance letter]

28 Oct 2024

PONE-D-23-27190R3 

PLOS ONE

Dear Dr. Foong, 

I'm pleased to inform you that your manuscript has been deemed suitable for publication in PLOS ONE. Congratulations! Your manuscript is now being handed over to our production team.

Kind regards, 

on behalf of

Dr. Musa Adekunle Ayanwale 

Academic Editor

PLOS ONE